# Improved Tensile Ductility by Severe Plastic Deformation for Nano-Structured Metallic Glass

**DOI:** 10.3390/ma12101611

**Published:** 2019-05-16

**Authors:** Yue Dong, Suya Liu, Johannes Biskupek, Qingping Cao, Xiaodong Wang, Jian-Zhong Jiang, Rainer Wunderlich, Hans-Jörg Fecht

**Affiliations:** 1Institute of Functional Nanosystems, University of Ulm, Albert-Einstein Allee 47, 89081 Ulm, Germany; yue.dong@uni-ulm.de (Y.D.); rainer.wunderlich@uni-ulm.de (R.W.); 2International Center for New-Structured Materials, State Key Laboratory of Silicon Materials and School of Materials Science and Engineering, Zhejiang University, Hangzhou 310027, China; suya.liu@kit.edu (S.L.); caoqp@zju.edu.cn (Q.C.); wangxd@zju.edu.cn (X.W.); 3Institute for Nanotechnology, Karlsruhe Institute of Technology, Hermann-von-Helmholtz-Platz 1, 76344 Eggenstein-Leopoldshafen, Germany; 4Electron Microscopy Group of Materials Science, University of Ulm, Albert-Einstein Allee 11, 89081 Ulm, Germany; johannes.biskupek@uni-ulm.de

**Keywords:** metallic glass, nano-heterogeneous structure, high-pressure torsion, tensile plasticity

## Abstract

The effect of severe plastic deformation by high-pressure torsion (HPT) on the structure and plastic tensile properties of two Zr-based bulk metallic glasses, Zr_55.7_Ni_10_Al_7_Cu_19_Co_8.3_ and Zr_64_Ni_10_Al_7_Cu_19_, was investigated. The compositions were chosen because, in TEM investigation, Zr_55.7_Ni_10_Al_7_Cu_19_Co_8.3_ exhibited nanoscale inhomogeneity, while Zr_64_Ni_10_Al_7_Cu_19_ appeared homogeneous on that length scale. The nanoscale inhomogeneity was expected to result in an increased plastic strain limit, as compared to the homogeneous material, which may be further increased by severe mechanical work. The as-cast materials exhibited 0.1% tensile plasticity for Zr_64_Ni_10_Al_7_Cu_19_ and Zr_55.7_Ni_10_Al_7_Cu_19_Co_8.3_. Following two rotations of HPT treatment, the tensile plastic strain was increased to 0.5% and 0.9%, respectively. Further testing was performed by X-ray diffraction and by differential scanning calorimetry. Following two rotations of HPT treatment, the initially fully amorphous Zr_55.7_Ni_10_Al_7_Cu_19_Co_8.3_ exhibited significantly increased free volume and a small volume fraction of nanocrystallites. A further increase in HPT rotation number did not result in an increase in plastic ductility of both alloys. Possible reasons for the different mechanical behavior of nanoscale heterogeneous Zr_55.7_Ni_10_Al_7_Cu_19_Co_8.3_ and homogeneous Zr_64_Ni_10_Al_7_Cu_19_ are presented.

## 1. Introduction

Metallic glasses (MGs), discovered in 1960 [1], are metallic bonded amorphous solids with no long-range translation/orientation order, formed by quenching from melt. Short-range topological and chemical orders on a length scale of 0.5–1 nm are often detected in MGs. More than two decades ago, multicomponent bulk metallic glass-forming alloys were synthesized [2,3,4,5,6,7,8,9]. Zirconium (Zr)-based MGs exhibit good glass-forming ability [2,7,8] and many outstanding properties such as high elastic limits, strength, and corrosion resistance [10,11,12,13,14,15,16]. Lacking a crystalline structure with dislocations, the mechanical deformation of MGs in tension is concentrated in a single or a few shear bands (SBs). Even though quite large ductility can be observed under constrained loading, such as in compression, tensile strain leads to catastrophic failure without plasticity, which restricts the potential application of MGs as structural materials [17,18].

Considerable efforts were devoted to improving the ductility of MGs. Creating more free volume and limiting the propagation of SBs are two key issues to improve tensile plasticity of MGs [19,20]. Recently, nano-structured glasses (nanoglasses), which are amorphous solids consisting of nanometer-sized glassy regions connected by a glass/glass interface, were reported [21,22,23]. Such glasses could have the following features: (1) denser glassy particles could hinder the propagation of SBs, acting as a second-phase strengthening mechanism; (2) an enhanced free volume, due to the misfit between the atoms at the boundaries between glassy particles, could offer more sites for a deformation-induced shear transformation zone (STZ), leading to the formation of more small SBs [21]. Computation simulations suggest that the heterogeneous free volume presented in the “glass-in-glass” structure could toughen the material and increase its ductility [24,25,26]. Nanoglasses exhibit unique atomic structure [21], electronic structure, thermal stability [27], and mechanical behavior [28]. Several methods were developed to produce the nanoscale heterogeneous MGs, including inert-gas condensation [21], severe plastic deformation [21], and magnetron sputtering [23]. In our previous investigation of the effect of the Co concentration on the structural and mechanical properties of a series of Zr-based MGs, a concentration was found, Zr_55.7_Ni_10_Al_7_Cu_19_Co_8.3_, which exhibited a high elastic strain limit and ductility in compression. The alloy exhibited nanoscale structural inhomogeneity resembling a phase separation, which could originate from the positive enthalpy of mixing of Cu and Co (∆H(Cu-Co) = +6 kJ/mol) [28]. However, its tensile plasticity is still poor, which needs to be improved if the material is to be used for industrial applications.

Compared with the matrix, SBs have dilated structure and less density. Previous studies [11,29] demonstrated that SBs induced by mechanical work in MGs lead to a free volume increase and interaction of SBs in the following compression/tension tests, resulting in the enhancement of ductility. High-pressure torsion (HPT) is a severe plastic deformation technique. It can achieve large plastic strain on MGs without material failing due to the geometric confinement [30,31]. A high density of SBs with excess free volume can be introduced. While all the HPT investigations with MGs were concerned with homogeneous materials [32,33,34,35,36,37,38,39,40,41,42,43,44,45], rare work of HPT testing of heterogeneous MGs is reported. Given the potential contribution of an inhomogeneous structure to the ductility of MGs in tension, we performed a comparative investigation of the HPT effect on the tensile ductility of two Zr-based MGs, Zr_55.7_Ni_10_Al_7_Cu_19_Co_8.3_ and Zr_64_Ni_10_Al_7_Cu_19_. The former exhibits a nanoscale structure and the latter does not. Hereafter, we designate Zr_55.7_Ni_10_Al_7_Cu_19_Co_8.3_ and Zr_64_Ni_10_Al_7_Cu_19_ as Zr-8Co and Zr-0Co, respectively.

## 2. Materials and Methods

Ingots with compositions of Zr_64_Ni_10_Al_7_Cu_19_ and Zr_55.7_Ni_10_Al_7_Cu_19_Co_8.3_ were prepared by alloying the pure elements in an arc melter in a high-purity Ar atmosphere. Large-scale homogeneity of the ingots was facilitated by remelting them five times and was verified by energy-dispersive X-ray spectroscopy analysis (Leo-1550, ZEISS, Oberkochen, Germany). Glassy materials were obtained by suction casting in a water-cooled copper mold with a circular shape 10 mm in diameter and 0.8 mm thick. Both sides of the samples were ground using 400, 800, 1200, and 2000 grit SiC papers to 0.5 mm thickness to remove the surface layer. Severe plastic deformation treatments by high-pressure torsion were performed on a custom-built HPT machine (A. Klement, Lang, Austria) under a quasi-hydrostatic pressure of 6 GPa, at room temperature using a rotation speed of 0.2 rpm. The rotation numbers, N, used here were 2, 5, 10, and 20 (N = 2, 5, 10, and 20). The introduced shear strain γ of different positions on the discs can be calculated through γ = 2πNR/h, where h is the thickness of the sample, and R is the distance from the sample center. In order to ensure the same strain history, the materials used for further studies were all cut from the same position on the HPT-treated discs at about 1–2 mm distance from the center.

A differential scanning calorimeter (DSC) (Perkin Elmer Pyris 1, Waltham, MA, USA) was used to analyze the thermal properties of all studied MGs. The heating range was 353–853 K with a heating rate of 20 K/min under argon flow of high purity. Two runs back to back were performed with each sample. Assuming that, after the first run, the sample was in a stable thermal state, the bias heat flow-corrected heat flow signal was obtained by subtracting the heat flow of the second run from that of the first.

Transmission electron microscopy (TEM) (aberration corrected FEI Titan 80/300 operating at 300 kV, Hillsboro, OR, USA) was applied for selected samples to monitor nanometer-scale structural changes in the studied MGs. Electron-transparent TEM specimens were produced by mechanical grinding and polishing followed by low-angle Ar-ion milling. The dark-field images were captured using some part of the first diffraction halo. The structure of all samples before and after HPT was also examined by Cu-Kα radiation X-ray diffraction (XRD) (Siemens D 5005 operating at 40 kV, Munich, Germany). To characterize atomic structures, synchrotron radiation-based XRD measurements using a wavelength of 0.2906 Å at HASYLAB/DESY in Hamburg were carried out for selected samples. After background subtraction and corrections for sample absorption, fluorescence, and Compton scattering, the structure factors S(Q) were obtained from the integrated intensity data.

Nanoindentation measurements (MTS NANO Indenter XP testing system, Oak Ridge, TN, USA) with a Berkovich indenter were performed on the well-polished BMG samples from their center to edge to analyze the hardness change. The maximum displacement was 2000 nm. Each sample was measured at least three times. Tensile testing was performed with a MicroDAC (Kammrath and Weiss) tensile test stage built in a high-resolution field-emission scanning electron microscope (SEM) (Leo-1550, ZEISS, Oberkochen, Germany) allowing in situ observation of the development of SBs and fracture. The as-cast samples and HPT samples (0.4 mm thickness) were polished to a thickness of 0.3 mm and to a mirror finish using SiC papers and diamond paste. In order to obtain reliable results, dog-bone shapes with a standard gauge size of 2 mm × 1 mm were cut from discs at the same position as other measured samples, 1–2 mm from the center, for tensile measurements. The strain rate used was 5 × 10^−5^ s^−1^ until fracture. The load–displacement curves and sample surface images were recorded during the test. It should be noted that, on the load-displacement curve, the initial deformation part is unreliable because of the deformation of the steel sample holder. The accurate elastic strain of MGs should be estimated by directly measuring the sample length change in SEM images during tension. All plastic deformation is localized on the thin dog-bone-shaped samples. Each measurement was performed twice with a different sample of the same preparation.

## 3. Results

### 3.1. Thermal Analysis

All samples investigated exhibited the typical DSC heat flow curves for amorphous samples with exothermal relaxation followed by glass transition and crystallization (Figure 1a,b). Zr-0Co MGs have one-step crystallization at temperature T_x1_ = 723 K and Zr-8Co MGs have two-step crystallization at lower (701 K) and higher (794 K) temperatures. The main qualitative features were similar for the as-cast and HPT-treated samples. In particular HPT treatment with N = 2, 5, 10, and 20 had no influence on the glass transition and crystallization temperatures within error limits. Principal numbers are shown in Table 1. Closer inspection of the thermal relaxation behavior before glass transition, shown in Figure 1c,d, revealed differences between the Zr-0Co and Zr-8Co. Details are listed in Table 2. It is apparent that, in both glasses, HPT (N = 2) introduced the highest exothermal enthalpy of relaxation. In addition, the enthalpy stored in the Zr-8Co glass was more than twice that introduced in the Zr-0Co glass.

### 3.2. Microstructure

#### 3.2.1. TEM

Figure 2a,b and Figure 3a,b show the dark-field TEM and bright-field high-resolution TEM (HRTEM) images of the as-cast Zr-0Co and Zr-8Co glassy alloys. The inserts show the electron diffraction patterns. Zr-0Co exhibited a relatively homogeneous structure on a length scale >2 nm, while Zr-8Co showed a clear contrast variation on a length scale of 5–10 nm indicative of a phase separation.

Figure 2c–f and Figure 3c–f show TEM images of the HPT-treated (N = 2 and 20) samples of the alloys Zr-0Co and Zr-8Co, respectively. For Zr-0Co, no significant difference of the microstructures of the as-cast and HPT-treated (N = 2 and 20) samples was observed. In contrast, in the Zr-8Co HPT-treated (N = 2) sample, some nanocrystallites were consistently observed, as shown in Figure 3c. The dark lines indicated in Figure 3c were interpreted as SBs. Apparently, more SBs were introduced in Zr-8Co as compared to the Zr-0Co alloy. More nanocrystalline particles were found along SBs as compared to the matrix, which could be explained by high-degree plastic deformation in SBs during HPT. Moreover, growth in the size of nanocrystalline particles was also observed on the Zr-8Co sample (diameter 5–10 nm, marked by arrows and circles in Figure 3e,f).

#### 3.2.2. XRD

Cu-Kα radiation and synchrotron radiation-based XRD patterns of the as-cast and HPT-treated alloys are shown in Figure 4 and Figure 5, respectively. For the as-cast and HPT-treated (N = 2, 5, 10, and 20) states, the XRD patterns of both compositions showed the typical amorphous diffraction patterns with a broad peak and no indication of crystallization. The width of the amorphous diffraction peaks (at half height) of both types of alloys increased after HPT. The corresponding numbers are shown in Table 2. The number density of the nanocrystallites in the HPT-treated Zr-8Co alloy samples was too low to show up as distinct diffraction peaks in the XRD patterns of the predominantly amorphous alloy.

### 3.3. Mechanical Properties

Since HPT can modify the structure in both homogeneous and nanometer-scale heterogeneous MGs, their mechanical behavior should be different. Hardness of the as-cast and HPT-treated samples was measured first. Zr-0Co samples had a nanoindentation hardness of 5.6 ± 0.2 GPa. For Zr-8Co MG, the value was higher (6.1 ± 0.2 GPa) due to the stronger Zr–Co atomic bonds. The HPT effect on hardness was analyzed in a previous publication [43]. In this work, we focus on the SB behavior in MGs during nanoindentation measurements.

The tension stress–strain curves of the as-cast and HPT-treated samples are shown in Figure 6. Similar to the results reported in many other publications [46,47], both as-cast MGs showed almost no plastic strain in tension (~0.1%). Unlike the compression process, under tensile load, SBs easily extend to cracks, which leads to catastrophic fracture very fast after yielding. After HPT (N = 2), the tensile ductility of both samples was improved. The plastic strain of the Zr-0Co MG increased to 0.5%. The improvement for the Zr-8Co MG was more significant, reaching 0.9%. This is a high value for MGs under tension with a slow strain rate at room temperature. For HPT treatment (N = 5, 10, and 20), the plastic strain of both types of MGs decreased to 0.1–0.3%.

## 4. Discussion

### 4.1. Evolution of Free Volume

The free volume model is one of the most commonly used for the description of properties of MGs [16,41,48,49,50]. As discussed above, MGs with more free volume often have more homogeneous deformation and better plasticity, which is caused by the formation of more STZs at sites with high free volume during deformation.

For Zr-based MGs [49], a correlation between free volume Δv_fv_ and structure relaxation enthalpy (ΔH)_fv_ before the glass transition during DSC measurements was proposed as (ΔH)_fv_=β ∙ Δv_fv_, where β is a constant. Table 2 lists the enthalpy of relaxation in all studied MGs as a function of rotation number. It is clear that, after two rotations of HPT, both types MGs had a higher enthalpy of relaxation as compared to the as-cast state. This increase was more significant for the HPT-treated (N = 2) Zr-8Co MG than the HPT-treated (N = 2) Zr-0Co MG, i.e., 6.4 J/g as compared to 3.5 J/g, respectively [49,50,51]. The observed first maximum shift on structure factor S(q) to lower q values (shown in the inserts in Figure 5) was equivalent to an increase in the average atomic distance. The relative increase of the mean atomic volume V can be estimated as {q_0_/q_N_}^3^ ≈ {V_N_/V_0_}, where q_0_ and q_N_ are the positions of the first maximum on S(q) of the as-cast and of the N-rotation HPT-treated samples, respectively [22,52]. From these numbers, an increase in mean volume per atom of ~0.3% and ~0.7% was obtained for HPT-treated (N = 2) Zr-0Co and Zr-8Co, respectively, indicating the creation of excess free volume. Furthermore, the diffraction peak halfwidth can also reflect the structure change of MGs [53]. One of the main reasons of peak broadening is the formation of an inhomogeneous amorphous structure. As shown in Table 2, the halfwidth of the XRD diffraction peak of the as-cast Zr-8Co MG (6.1°) was larger than that of the Zr-0Co MG (5.4°), due to the nano-heterogeneous glassy structure in Zr-8Co as compared with Zr-0Co. Following two rotations of HPT treatment, the peak width of both types of alloys increased, which varied from 6.1° to 6.8° and 5.4° to 5.9° for the abovementioned compositions, respectively. As for the reason for the XRD peak broadening, due to the small content of the nanocrystalline particles (0–5%, as detected by TEM and DSC) induced by HPT treatment, this peak broadening mainly resulted from the inhomogeneity caused by shear band formation. The increase in atomic disorder is associated with an increase in the mean volume per atom and localized free volume. As such, the different measures of disorder were investigated on a similar scale for both types of alloys. From these results, it can be concluded that the free volume of both types of MGs increased following two rotations of HPT. Furthermore, the free volume increase of the treated nano-heterogeneous Zr-8Co sample is higher than that of the homogeneous Zr-0Co sample after HPT (N = 2). The reason is that atomic misfit at the boundaries between different glassy phases could offer more sites for deformation-induced STZ [21]. Thus, more SBs with excess free volume can be formed in nanostructured MGs during plastic deformation, which was observed and discussed in our previous publication [28].

By further increasing the rotation number (N = 5, 10, and 20), the enthalpy of relaxation in both types of MGs decreased compared with the HPT treatment (N = 2) as shown in Table 2, which could be caused by deformation-accelerated diffusion and structure relaxation during HPT (N = 5, 10, and 20) [54]. This process is similar to the defect content evolution in alloys with time during ball milling [55,56]. The deformation-induced defect (defect creation) and deformation-accelerated diffusion (defect annihilation) during ball milling are both present, and they compete with each other during ball milling. Usually, at the initial stage, the deformation-induced defect is dominant, while, at the later stage, deformation-accelerated diffusion becomes important; finally, a balance is often reached. Although the HPT treatment is not exactly the same as ball milling, a similar deformation-induced defect and deformation-accelerated diffusion should also occur. After HPT (N = 2), more SBs (could be treated as defects) were formed, i.e., the deformation-induced defect process was dominant. Thus, the enthalpy of relaxation increased in both types of MGs after HPT (N = 2). With further rotation, deformation-accelerated diffusion became important and structure relaxation occurred, resulting in the reduction of the enthalpy of relaxation and the free volume. In fact, the sample temperature enhancement during severe plastic deformation further supports our view. As previously calculated [32,33], the temperature in the Zr-based MG samples will reach their T_g_ after 2–5 rotations of HPT. Thus, for more rotations, structural relaxation during the treatment becomes non-negligible, resulting in the reduction of structure relaxation enthalpy in the DSC results. Furthermore, the growth of nanometer-sized crystalline particles and the homogenization of the amorphous matrix in the Zr-8Co sample (Figure 3e,f) led to this reduction being more significant than in HPT-treated Zr-0Co samples (N = 5, 10 and 20).

### 4.2. Behavior of Shear Bands

HPT introduced some change of the free volume content in both MGs. In order to investigate its effect on SB formation in tension, in situ SEM was used to observe the SB initiation and evolution on the as-cast and HPT-treated Zr-0Co and Zr-8Co MGs samples. The polished surfaces were smooth except for a few scratches. Figure 7 and Figure 8 show the SEM micrographs of Zr-0Co and Zr-8Co samples just before the yielding point. Figure 7a shows that one major SB was formed, cutting through the sample, and several smaller SBs beside it were observed for the as-cast Zr-0Co MG. On the surface of the HPT-treated (N = 2) Zr-0Co sample (Figure 7b), multiple SBs appeared under yield stress perpendicular to the tension orientation because of the increased free volume. This phenomenon is similar to the tensile test on rolled MG samples [11]. The SBs introduced by suitable plastic deformation might not only lead to the increased free volume, but also tolerance of the applied strain more homogeneously across the deformed region, thereby avoiding catastrophic failure. Upon further increasing the HPT deformation degree (up to N = 20), the number of tension-induced SBs was reduced (Figure 7c) due to the free volume annihilation as discussed above. For the as-cast Zr-8Co sample, more than one major SBs were detected, most likely due to the inhibition effect of the nanometer-scale glassy second phase during SBs propagation. On the surface of the HPT-treated (N = 2) Zr-8Co sample, multiple SBs were easily observed (Figure 8b) due to the high number of initiation sites in this glassy structure with the highest free volume content. Similar to Zr-0Co MGs, only two to three SBs could be observed on the surface of the HPT-treated (N = 20) Zr-8Co sample under yield stress due to the reduction of free volume, as shown in Figure 8c.

During tensile tests, a large number of SBs formed on both N = 2 HPT-treated MGs, although the SB morphology on both types of samples was different (Figure 7b and Figure 8b). Finite element analysis of the tensile tests was done by Mo Li et al. [26]. They used Gaussian and bimodal distribution of free volume to simulate the spatial heterogeneity in MGs. The calculation results showed that bimodal-like free volume dispersion can effectively improve the sample plasticity. In the HPT-treated (N = 2) Zr-8Co MG, free volume content was high in the introduced SBs and the formed nanocrystalline particles had little free volume. In this heterogeneous structure, initial deformation started at the soft mesh regions and continued to exist around their original location. As the deformation continued, the deformation regions were still restricted to their original location, although more new deformed regions were created elsewhere. When localized regions finally developed, the deformation bands did not look smooth and straight, but rather appeared rugged and zigzag with many side bands (as shown in Figure 8b and Appendix A (Appendix A). Moreover, the localized deformation zones were spread more widely and no through shear band across the sample formed at larger deformation [26]. Upon further increasing the number of HPT rotations (N = 5, 10, 20), the free volume dispersion in the samples was homogenized.

Nanometer-sized crystalline particles (3–10 nm) formed and grew in the nanoscale heterogeneous MGs during HPT, which could not be observed in the homogeneous MG (Figure 3). The reason is that nanostructured Zr-8Co MG has less thermal stability (T_x1_ − T_g_). As shown in Table 1, (T_x_ − T_g_) of the Zr-8Co sample is only 26 K, while it is 72 K for the Zr-0Co sample. For crystalline composite materials, the nanometer scale reinforcements can hinder the sliding of dislocations. To study their effect on the mechanical properties of MGs, nanoindentaion measurement results of the HPT-treated (N = 2) Zr-0Co and Zr-8Co MGs were analyzed. The two smooth curves shown in Figure 9a illustrate that both types of MGs had relatively homogeneous deformation. If we focus on the beginning of plastic deformation (Figure 9b), some small “serrations” with a sudden load increase could be observed for the Zr-8Co sample. The reason is that the initiated SB was arrested by the nanocrystalline particles. With the increase of load, new SBs formed and the deformation continued. This phenomenon could not be found on the treated Zr-0Co sample. Instead, some flat “steps” appeared on its nanoindentaion curve, indicating no obstacle for SBs propagation. This can be further confirmed by the velocity profiles of the indenter, as shown in Figure 9c,d. Abruptly deceleration and acceleration of the indenter could be seen on the velocity–displacement curve of the HPT-treated (N = 2) Zr-8Co MG at the corresponding displacements of the “serrations” on Figure 9b, indicating the displacement burst due to new SB activity followed by an increase in load. On the curve of the HPT-treated (N = 2) Zr-0Co MG, the indenter velocity change was relatively uniform. On the SEM images (Figure 9e,f), more SBs could be observed around the indentation on the HPT-treated (N = 2) Zr-8Co MG.

### 4.3. Propagation of Cracks

In alloys, the second-phase reinforcements with suitable size can blunt cracks, resulting in better mechanical performance. Figure 10 illustrates the SEM images for crack tips on the HPT-treated (N = 2) Zr-0Co and Zr-8Co samples just before fracture. It is clear that, for the HPT-treated (N = 2) Zr-8Co sample, many SBs branching and interacting with each other at the tip could be observed, which were not detected for the HPT-treated (N = 2) Zr-0Co sample. In the HPT-treated (N = 2) Zr-8Co sample, the nanometer-scale crystalline second phase blocked and/or blunted the propagation of the crack. The stress was released by the formation of new SBs. The propagation of the newly formed SBs was inhibited again by branching and intersection [12,57].

The fracture surface morphologies of the HPT-treated (N = 2) Zr-0Co and Zr-8Co MGs are shown in Figure 11. The typical river-like features can be observed on both samples. It is worth noting that a flat and feature-less region runs parallel to the fractured edge. This corresponds to the shear offset caused by the SB propagation. The width of the shear offset was larger in the Zr-8Co sample than that in the Zr-0Co sample. This means that SBs in the HPT-treated (N = 2) Zr-8Co MG had less susceptibility to becoming a shear crack and each of them could carry a larger plastic strain. In combination with the increased number of SBs as analyzed above, the good tensile ductility of the HPT-treated (N = 2) Zr-8Co MG can be explained [11,58].

### 4.4. HPT Effects on Tensile Ductility of Both Types of MGs

Two rotations of HPT treatment showed the best enhancement effect on tensile ductility for both types of MGs, as shown on the stress–strain curves (Figure 6). The correlation of the tensile plastic strain with the structure relaxation enthalpy on heating below the glass transition and the position of the first maximum in the S(q) indicated that the free volume played an important role in the increased plasticity. After HPT for two rotations, multiple SBs with excess free volume were formed in both types of MGs, in which more STZs and even more SBs could be further induced following tension, resulting in more homogeneous plastic deformation. In addition, these SBs could also interact with each other, which could further retard their propagation, improving tensile ductility [59]. Further increasing the rotation number (N = 5, 10, and 20) led to the reduction of free volume (as discussed in Section 4.1) and the decrease of SBs formed in following tension (as discussed in Section 4.2), deteriorating the tensile ductility of samples.

After two rotations of HPT treatment, the nano-heterogeneous Zr-8Co sample had a more significant tensile plastic strain increase (0.1% to 0.9%) than the homogeneous Zr-0Co MG (0.1% to 0.5%). The first reason is that more SBs with excess free volume formed in the nano-heterogeneous structure during two rotations of HPT, illustrated by the change in structure relaxation enthalpy and mean atomic volume (Table 2). Thus, the increased SB initiation regions resulted in more homogeneous plastic deformation in the following tensile test. The second reason is the formation of nanocrystalline particles, which could effectively inhibit the propagation of SBs and cracks. By further increasing the HPT rotation number (N = 5, 10, and 20), plastic strain reduction was also more significant for the nanostructured Zr-8Co MG (0.9% to 0.2%), because of the larger free volume annihilation (Table 2) and growth of crystalline particles (Figure 3e,f). The crystalline particles have little pinning effect when their size is larger than the SB thickness (~10 nm) [16], whereby no displacement burst could be observed on the velocity profile of the nano-indenter on the Zr-Co8 (N = 20) sample. Their inhibition effect on crack propagation was also weakened. On the tip of the cracks, only very small SBs formed to release the stress concentration. Furthermore, their brittleness was harmful to the sample ductility.

## 5. Conclusions

In this work, the changes of microstructure and mechanical properties of the homogeneous Zr_64_Ni_10_Al_7_Cu_19_ MG (Zr-0Co) and nanometer-scale heterogeneous Zr_55.7_Ni_10_Al_7_Cu_19_Co_8.3_ MG (Zr-8Co) as a function of severe plastic deformation (HPT) were investigated.

It was found that the homogeneous structure of Zr-0Co MG remained fully amorphous during HPT up to 20 rotations. Nanocrystalline particles appeared in the Zr-8Co MG because of the lower thermal stability. Both samples had increased free volume after HPT for two rotations, which was more significant for Zr-8Co MG. Plasticity of both MG samples was improved by two rotations of HPT. A value of 0.9% plastic tensile strain was achieved on the HPT-treated (N = 2) Zr-8Co sample, which resulted from the higher number of formed SBs during tension and blunting of cracks by the nanocrystalline second phase. Further increasing the HPT rotation number (N = 5, 10, and 20) reduced sample plasticity.

## Figures and Tables

**Figure 1 materials-12-01611-f001:**
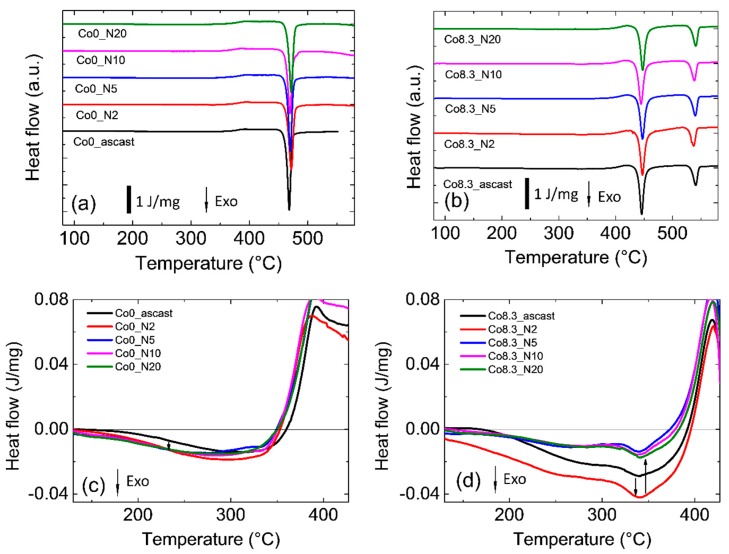
Differential scanning calorimeter curves of (**a**) Zr_64_Ni_10_Al_7_Cu_19_ (Zr-0Co) and (**b**) Zr_55.7_Ni_10_Al_7_Cu_19_Co_8.3_ (Zr-8Co) metallic glasses (MGs) as a function of the number of high-pressure torsion (HPT) rotations. Relaxation energy release below T_g_ of (**c**) Zr-0Co and (**d**) Zr-8Co MGs as a function of the number of HPT rotations.

**Figure 2 materials-12-01611-f002:**
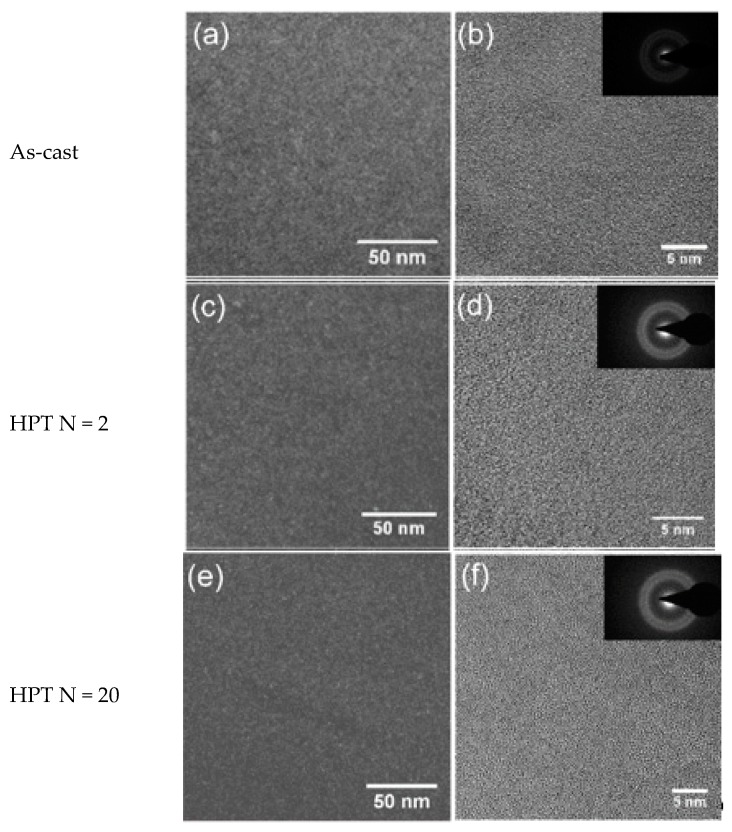
Transmission electron microscope (TEM) images of Zr-0Co. (**a**) Dark-field image of as-cast sample. (**b**) High-resolution TEM (HRTEM) image of as-cast sample. (**c**) Dark-field image of the HPT-treated (N = 2) sample. (**d**) HRTEM image of the HPT-treated (N = 2) sample. (**e**) Dark-field image of the HPT-treated (N = 20) sample. (**f**) HRTEM image of the HPT-treated (N = 20) sample. The inserts show the electron diffraction patterns.

**Figure 3 materials-12-01611-f003:**
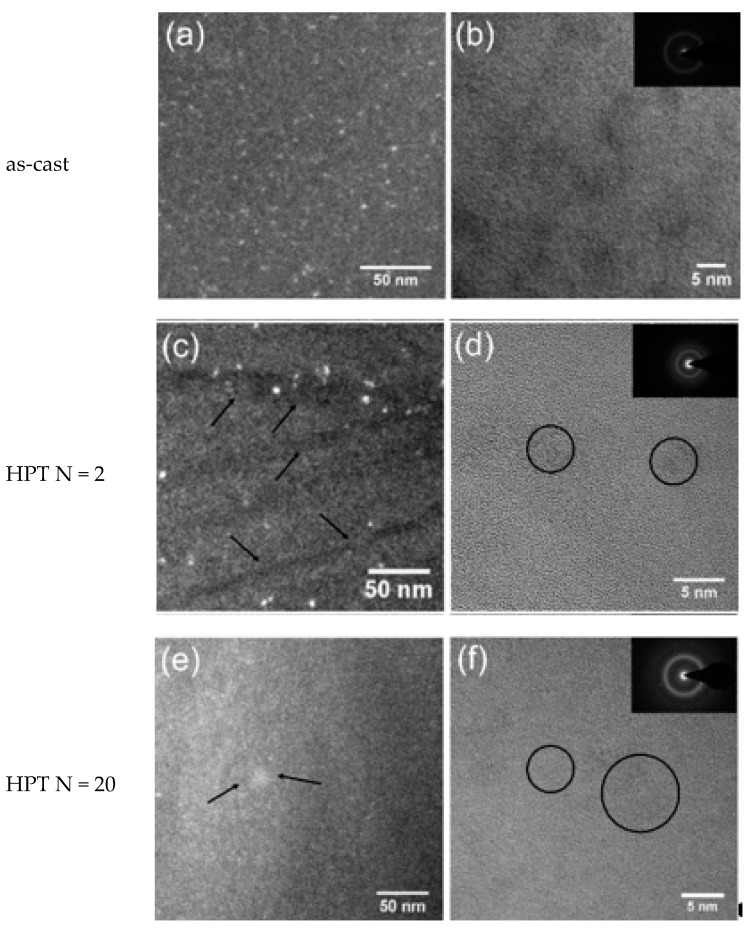
Transmission electron microscope images of Zr-8Co. (**a**) Dark-field image of as-cast sample. (**b**) HRTEM image of as-cast sample. (**c**) Dark-field image of the HPT-treated (N = 2) sample. The arrows indicate the shear bands. (**d**) HRTEM image of the HPT-treated (N = 2) sample. The circles indicate the nanocrystalline particles. (**e**) Dark-field image of the HPT-treated (N = 20) sample. The arrows indicate the nanocrystalline particles. (**f**) HRTEM image of the HPT-treated (N = 20) sample. The circles indicate the nanocrystalline particles. The inserts show the electron diffraction patterns.

**Figure 4 materials-12-01611-f004:**
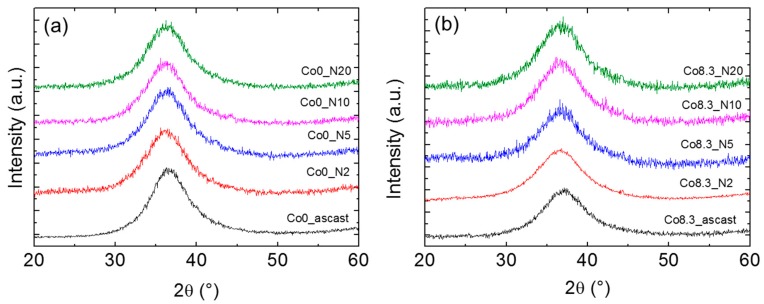
X-ray diffraction (XRD) patterns of as-cast MGs and MG samples after HPT treatments: (**a**) Zr-0Co and (**b**) Zr-8Co.

**Figure 5 materials-12-01611-f005:**
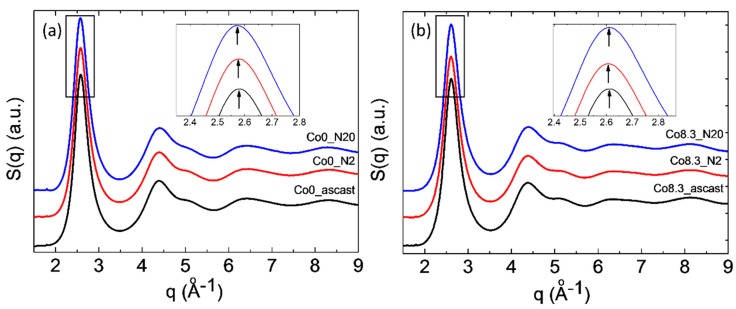
Structure factor S(q) of as-cast MGs and MG samples after HPT treatments: (**a**) Zr-0Co and (**b**) Zr-8Co.

**Figure 6 materials-12-01611-f006:**
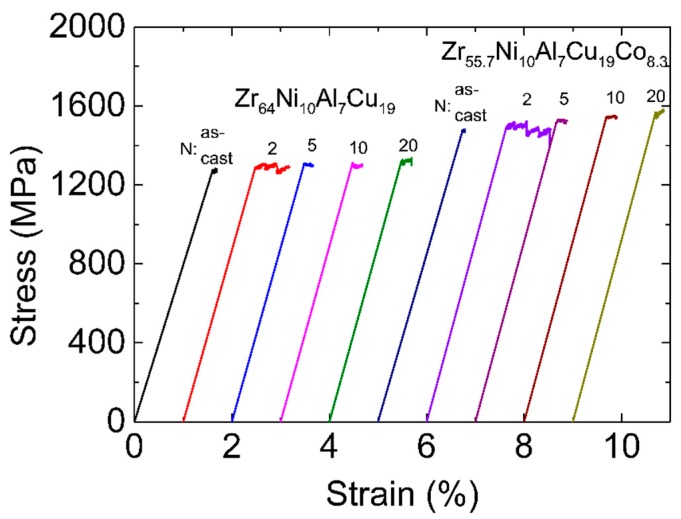
Tensile stress–strain curves of Zr-0Co and Zr-8Co samples before and after HPT processes. The curves are shifted relative to each other for clarity.

**Figure 7 materials-12-01611-f007:**
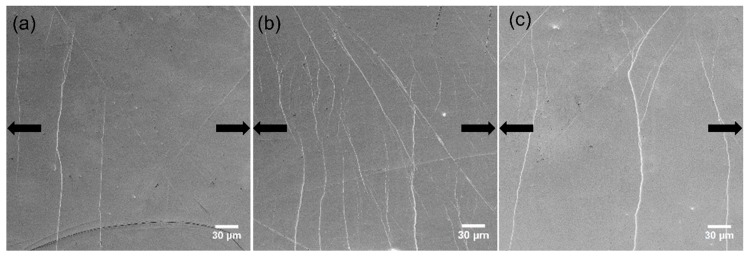
Densest areas of shear bands on the surface of the Zr-0Co samples (**a**) as-cast, (**b**) N = 2, and (**c**) N = 20, when the load almost reached the yield point. The arrows indicate the tensile orientation.

**Figure 8 materials-12-01611-f008:**
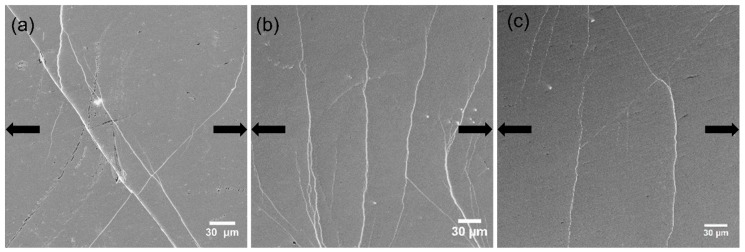
Densest areas of shear bands on the surface of Zr-8Co samples (**a**) as-cast, (**b**) N = 2, and (**c**) N = 20, when the load almost reached the yield point. The arrows indicate the tensile orientation.

**Figure 9 materials-12-01611-f009:**
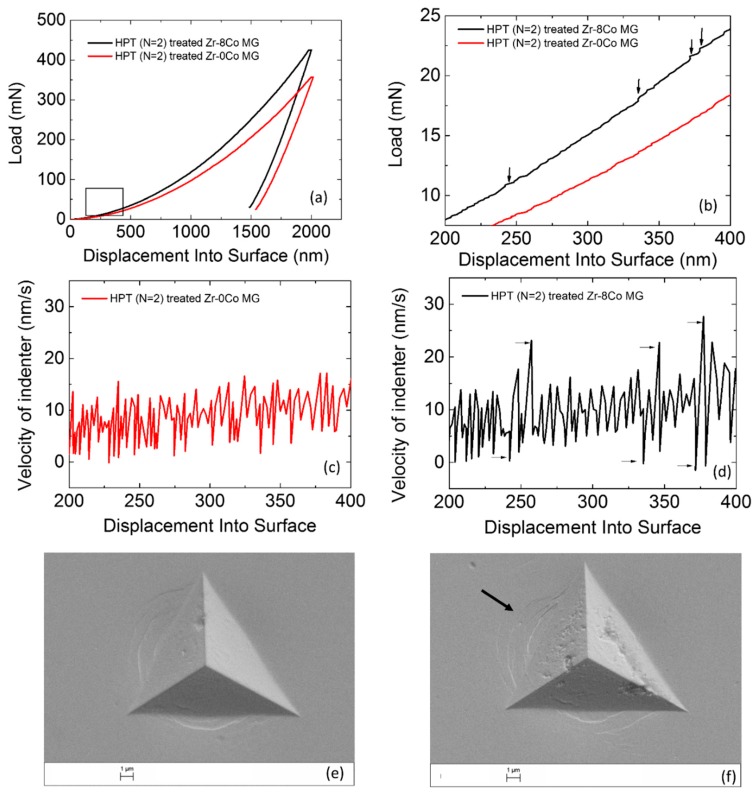
(**a**) Nanoindentation measurement curves of the HPT-treated (N = 2) Zr-0Co MG and HPT-treated (N = 2) Zr-8Co MG. (**b**) Beginning of plastic deformation on nanoindentation measurement curves of both types of MGs. (**c**) Velocity profile of the indenter on the HPT-treated (N = 2) Zr-0Co MG. (**d**) Velocity profile of the indenter on the HPT-treated (N = 2) Zr-8Co MG. (**e**) SEM image of indentation on the HPT-treated (N = 2) Zr-0Co MG. (**f**) SEM image of indentation on the HPT-treated (N = 2) Zr-8Co MG.

**Figure 10 materials-12-01611-f010:**
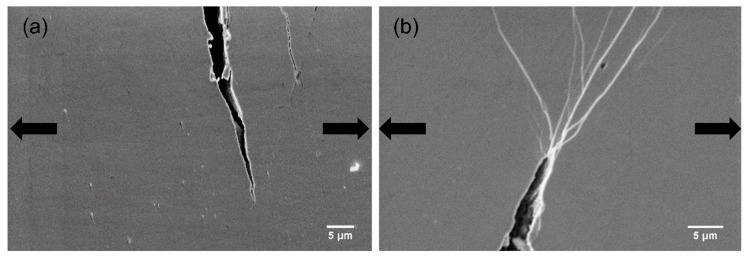
SEM images of cracks tips during tensile tests of the deformed HPT-treated (N = 2) samples of (**a**) Zr-0Co and (**b**) Zr-8Co. The arrows indicate the loading orientation.

**Figure 11 materials-12-01611-f011:**
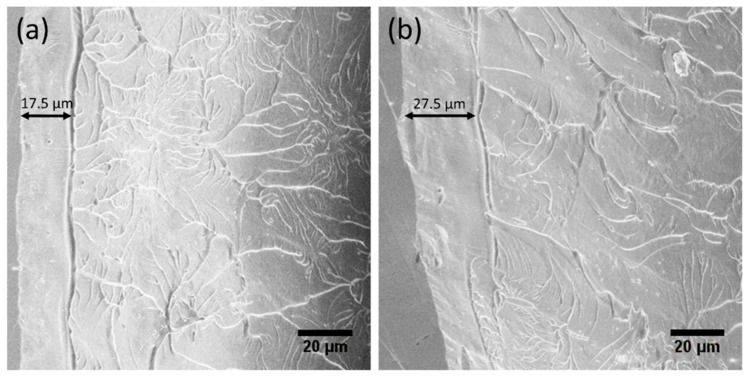
SEM fractographic images of (**a**) the HPT-treated (N = 2) Zr-0Co MG, and (**b**) the HPT-treated (N = 2) Zr-8Co MG.

**Table 1 materials-12-01611-t001:** Thermal parameters of glass transition temperature T_g_, first crystallization temperature T_x1_, second crystallization temperature T_x2_, melting temperature Tl, first crystallization enthalpy ∆H_x1_, and second crystallization enthalpy ∆H_x2_ of Zr_64_Ni_10_Al_7_Cu_19_ (Zr-0Co) and Zr_55.7_Ni_10_Al_7_Cu_19_Co_8.3_ (Zr-8Co) metallic glass (MG) samples before and after high-pressure torsion (HPT) processes (N = 2, 5, 10, and 20).

	Zr_64_Ni_10_Al_7_Cu_19_					Zr_55.7_Ni_10_Al_7_Cu_19_Co_8.3_				
	As Cast	N = 2	N = 5	N = 10	N = 20	As Cast	N = 2	N = 5	N = 10	N = 20
T_g_ (K)	648 ± 3	649 ± 3	647 ± 3	650 ± 3	648 ± 3	674 ± 3	676 ± 3	673 ± 3	675 ± 3	675 ± 3
T_x1_ (K)	720 ± 3	723 ± 3	724 ± 3	721 ± 3	723 ± 3	700 ± 3	699 ± 3	703 ± 3	701 ± 3	702 ± 3
T_x2_ (K)	-	-	-	-	-	794 ± 3	792 ± 3	795 ± 3	796 ± 3	794 ± 3
T_l_ (K)	1182 ± 3	1185 ± 3	1184 ± 3	1186 ± 3	1187 ± 3	1268 ± 3	1265 ± 3	1267 ± 3	1270 ± 3	1268 ± 3
∆H_x1_ (J/g)	−60 ± 2	−60 ± 2	−57 ± 2	−57 ± 2	−58 ± 2	−28 ± 2	−29 ± 2	−26 ± 2	−27 ± 2	−26 ± 2
∆H_x2_ (J/g)	-	-	-	-	-	−13 ± 1	−13 ± 1	−13 ± 1	−12 ± 1	−13 ± 1

**Table 2 materials-12-01611-t002:** Tensile plastic strain, enthalpy of relaxation, position of the first maximum on structure factor curves, and the width of X-ray diffraction (XRD) peak at half height of Zr-0Co and Zr-8Co MG samples before and after HPT processes (N = 2, 5, 10 and 20).

	Zr_64_Ni_10_Al_7_Cu_19_					Zr_55.7_Ni_10_Al_7_Cu_19_Co_8.3_				
	As Cast	N = 2	N = 5	N = 10	N = 20	As Cast	N = 2	N = 5	N = 10	N = 20
Tensile plastic strain (%)	0.1 ± 0.1	0.5 ± 0.1	0.2 ± 0.1	0.2 ± 0.1	0.2 ± 0.1	0.1 ± 0.1	0.9 ± 0.1	0.2 ± 0.1	0.1 ± 0.1	0.2 ± 0.1
Enthalpy of relaxation (J/g)	−4.9 ± 0.5	−8.4 ± 0.5	−7.1 ± 0.5	−7.5 ± 0.5	−7.3 ± 0.5	−10.9 ± 0.5	−17.3 ± 0.5	−6.0 ± 0.5	−6.2 ± 0.5	−6.3 ± 0.5
Position of first maximum (Å^−1^)	2.587 ± 0.001	2.584 ± 0.001	-	-	2.583 ± 0.001	2.620 ± 0.001	2.614 ± 0.001	-	-	2.617 ± 0.001
Width of XRD peak (°)	5.4 ± 0.1	5.9 ± 0.1	5.8 ± 0.1	5.7 ± 0.1	5.8 ± 0.1	6.1 ± 0.1	6.8 ± 0.1	6.8 ± 0.1	6.7 ± 0.1	6.7 ± 0.1

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
