# Peer review of "Improved Tensile Ductility by Severe Plastic Deformation for Nano-Structured Metallic Glass"

_materials, 2019, doi:10.3390/ma12101611_

Round 1
Reviewer 1 Report
The manuscript “Improved tensile ductility by severe plastic deformation for a nano-structured metallic glass” is devoted to study of tensile ductility of Zr-based bulk metallic glasses after SPD treatment. New interesting results have been obtained.
However, the work needs some correction. The authors compare the crystallization process and mechanical properties in a homogeneous and inhomogeneous amorphous phase. They also analyze the change in free volume after SPD in a homogeneous and inhomogeneous amorphous alloy. Different homogeneity depends on the chemical composition and is observed immediately after the preparation of amorphous samples. The increase in free volume after SPD undoubtedly follows from the shift of the maximum on the XRD towards smaller values of the wave vector. As for the change in the halfwidth of the diffuse peak, the analysis of its broadening seems incomplete. The broadening of the diffuse peak may be due to a number of reasons. One of the main reasons of halfwidth increase is the formation of an inhomogeneous amorphous structure. Both the parameters of the forming nanocrystalline structure and the properties of the sample depend on whether the crystallization process develops in the homogeneous or inhomogeneous amorphous phase (see, for example .A.Aronin et al. J.All. Comp. 715 (2017) 176). In a comparative analysis of the properties of a homogeneous and inhomogeneous amorphous sample, the possibility of forming a heterogeneous structure from an initially homogeneous amorphous phase should be necessarily discussed.
Author Response
Response to Reviewer 1 Comments
Point 1:
The authors compare the crystallization process and mechanical properties in a homogeneous and inhomogeneous amorphous phase. They also analyze the change in free volume after SPD in a homogeneous and inhomogeneous amorphous alloy. Different homogeneity depends on the chemical composition and is observed immediately after the preparation of amorphous samples. The increase in free volume after SPD undoubtedly follows from the shift of the maximum on the XRD towards smaller values of the wave vector. As for the change in the halfwidth of the diffuse peak, the analysis of its broadening seems incomplete. The broadening of the diffuse peak may be due to a number of reasons. One of the main reasons of halfwidth increase is the formation of an inhomogeneous amorphous structure. Both the parameters of the forming nanocrystalline structure and the properties of the sample depend on whether the crystallization process develops in the homogeneous or inhomogeneous amorphous phase (see, for example .A.Aronin et al. J.All. Comp. 715 (2017) 176). In a comparative analysis of the properties of a homogeneous and inhomogeneous amorphous sample, the possibility of forming a heterogeneous structure from an initially homogeneous amorphous phase should be necessarily discussed.
Response 1:
The authors sincerely thank the reviewer’s comments. We discussed more about the diffraction peak broadening phenomenon in 4.1. section of revised manuscript, which has been highlighted.
The diffraction peak halfwidth can reflect the structure change of MGs, as mentioned by the reviewer. One of the main reasons of peak broadening is the formation of an inhomogeneous amorphous structure. As shown in Table 2 the halfwidth of the XRD diffraction peak of as cast Zr-8Co MG (6.1°) is larger than Zr-0Co MG (5.4°), due to the nano-heterogeneous glassy structure in Zr-8Co as compared with Zr-0Co. By two rotations HPT treatment, the peak width of both-type alloys increases, which varies from 6.1° to 6.8° and 5.4° to 5.9° for the above-mentioned two compositions, respectively. As for the reason to the XRD peak broadening, due to the small content of the nanocrystalline particles (0%~5%, as detected by TEM and DSC) induced by HPT treatment, this peak broadening is mainly resulted from the inhomogeneity caused by shear band formation. In the revised manuscript, necessary references and discussion have been added accordingly.
Reviewer 2 Report
@page { margin: 0.79in } p { margin-bottom: 0.1in; line-height: 120% }
The manuscript is an experimental work on HPT deformed metallic glass samples. The authors use several characterization methods and successfully pinpoint the minor differences between the deformed and the as-cast states of two metallic glass samples. Moreover, they have identified an optimal HPT deformation in which some tensile plasticity can be induced. The text is well written, but it lengthy and contains some strange expressions. Focusing the text and minor polishing of the English would improve the manuscript.
I have the following problems with the manuscript.
lines 66-74: Last part of the Introduction can be rewritten to help the reader by simplifying the composite sentences.
line 77: "... in high purity Ar atmosphere ..."
lines 77-78 I would write: “Large scale homogeneity of the ingots was facilitate by remelting them 5 times and was verified ... “
lines 86-87: N is already defined in line 84
lines 112-113: tension sample dimensions are defined in line 88
BF image description is missing from the Methods. as it is not straightforward in an amorphous material. I resume it was captured using some part of the first halo.
Line 106: “… displacement was 2000nm”
lines 122-128: Fig 1a and b are not cited in the text.
Line 124: the exo peak is indicated as Tx1 in Table 1, please use this notation here as well.
Line 140, Table 1: The crystallization enthalpies are not mentioned in the text.
Line 161 and 166: Optionally, the authors may indicate in the different rows of Fig 2 and 3 the corresponding states of the material (as-cast, HPT N=2, HPT N=20)
line 184: Fig 5 is cited in the text after Fig 6
line 217: It is difficult to see the shift of the halo in the insert of Fig5
line 257 “HPT introduced some change of the free volume content in both MGs. In order to investigate its effect on SB formation in tension, …”
line 276: “During tensile tests, large number of SBs formed on both N=2 HPT treated Mgs.”
line 282-285: please simplify these sentences
line 312: “… new SB activity ...”
Optionally, Fig 12 can be omitted to reduce the length of the paper.
Please check the references.
Author names are not correct in Ref 33 and there is a typo in Ref 34 as well.
Author Response
Response to Reviewer 2 Comments
Points:
lines 66-74: Last part of the Introduction can be rewritten to help the reader by simplifying the composite sentences.
line 77: "... in high purity Ar atmosphere ..."
lines 77-78 I would write: “Large scale homogeneity of the ingots was facilitate by remelting them 5 times and was verified ... “
lines 86-87: N is already defined in line 84
lines 112-113: tension sample dimensions are defined in line 88
BF image description is missing from the Methods. as it is not straightforward in an amorphous material. I resume it was captured using some part of the first halo.
Line 106: “… displacement was 2000nm”
lines 122-128: Fig 1a and b are not cited in the text.
Line 124: the exo peak is indicated as Tx1 in Table 1, please use this notation here as well.
Line 140, Table 1: The crystallization enthalpies are not mentioned in the text.
Line 161 and 166: Optionally, the authors may indicate in the different rows of Fig 2 and 3 the corresponding states of the material (as-cast, HPT N=2, HPT N=20)
line 184: Fig 5 is cited in the text after Fig 6
line 217: It is difficult to see the shift of the halo in the insert of Fig5
line 257 “HPT introduced some change of the free volume content in both MGs. In order to investigate its effect on SB formation in tension, …”
line 276: “During tensile tests, large number of SBs formed on both N=2 HPT treated Mgs.”
line 282-285: please simplify these sentences
line 312: “… new SB activity ...”
Optionally, Fig 12 can be omitted to reduce the length of the paper.
Please check the references.
Author names are not correct in Ref 33 and there is a typo in Ref 34 as well.
Response:
The authors sincerely thank the reviewer’s comments. According to the suggestions of the reviewer, all the required modifications have been made and highlighted in the revised manuscript.
lines 66-74: The sentences have been simplified.
line 77: The sentence has been modified.
lines 77-78: The sentence has been modified.
lines 86-87: The sentence has been modified.
lines 112-113: The sentence has been modified.
Line 97: The dark field images were captured using some part of the first diffraction halo.
Line 106: The sentence has been modified.
lines 122-128: Fig 1a and b are cited in line 123.
Line 124: The sentence has been modified.
Line 140, Table 1: The crystallization enthalpies are used to prove the small content of crystalline particles in the following.
Line 161 and 166: Corresponding states of the material (as-cast, HPT N=2, HPT N=20) have been indicated on Fig.2 and Fig.3
line 184: Fig 5 is cited in line 175
line 217: The picture is improved and the values are given in Table2.
line 257 The sentence has been modified.
line 276: The sentence has been modified.
line 282-285: Sentences have been simplified.
line 312: The sentence has been modified.
Fig 12 has been omitted.
The references have been checked.
Reviewer 3 Report
In my opinion the results presented in the manuscript are new and valuable for understanding the potential mechanical properties of the glass-nanocrystalline composites. Presently the subject is slightly theoretical as the HPT method supplies very small samples, but who knows in future. Also the presentaion is good both from the point of language and redaction.
Author Response
Response to Reviewer 3 Comments
Response:
The authors sincerely thank the reviewer's positive comments.